# Future Modulation of Gut Microbiota: From Eubiotics to FMT, Engineered Bacteria, and Phage Therapy

**DOI:** 10.3390/antibiotics12050868

**Published:** 2023-05-08

**Authors:** Carlo Airola, Andrea Severino, Serena Porcari, William Fusco, Benjamin H. Mullish, Antonio Gasbarrini, Giovanni Cammarota, Francesca Romana Ponziani, Gianluca Ianiro

**Affiliations:** 1Digestive Disease Center, Fondazione Policlinico Universitario Agostino Gemelli IRCCS, 00168 Rome, Italy; 2Dipartimento Universitario di Medicina e Chirurgia Traslazionale, Università Cattolica del Sacro Cuore, 00168 Rome, Italy; 3Division of Digestive Diseases, Department of Metabolism, Digestion and Reproduction, Faculty of Medicine, St Mary’s Hospital Campus, Imperial College London, London W2 1NY, UK; b.mullish@imperial.ac.uk; 4Departments of Gastroenterology and Hepatology, St Mary’s Hospital, Imperial College Healthcare NHS Trust, London W2 1NY, UK

**Keywords:** eubiotics, fecal microbiota transplant, phage therapy, microbiota, rifaximin, dysbiosis

## Abstract

The human gut is inhabited by a multitude of bacteria, yeasts, and viruses. A dynamic balance among these microorganisms is associated with the well-being of the human being, and a large body of evidence supports a role of dysbiosis in the pathogenesis of several diseases. Given the importance of the gut microbiota in the preservation of human health, probiotics, prebiotics, synbiotics, and postbiotics have been classically used as strategies to modulate the gut microbiota and achieve beneficial effects for the host. Nonetheless, several molecules not typically included in these categories have demonstrated a role in restoring the equilibrium among the components of the gut microbiota. Among these, rifaximin, as well as other antimicrobial drugs, such as triclosan, or natural compounds (including evodiamine and polyphenols) have common pleiotropic characteristics. On one hand, they suppress the growth of dangerous bacteria while promoting beneficial bacteria in the gut microbiota. On the other hand, they contribute to the regulation of the immune response in the case of dysbiosis by directly influencing the immune system and epithelial cells or by inducing the gut bacteria to produce immune-modulatory compounds, such as short-chain fatty acids. Fecal microbiota transplantation (FMT) has also been investigated as a procedure to restore the equilibrium of the gut microbiota and has shown benefits in many diseases, including inflammatory bowel disease, chronic liver disorders, and extraintestinal autoimmune conditions. One of the most significant limits of the current techniques used to modulate the gut microbiota is the lack of tools that can precisely modulate specific members of complex microbial communities. Novel approaches, including the use of engineered probiotic bacteria or bacteriophage-based therapy, have recently appeared as promising strategies to provide targeted and tailored therapeutic modulation of the gut microbiota, but their role in clinical practice has yet to be clarified. The aim of this review is to discuss the most recently introduced innovations in the field of therapeutic microbiome modulation.

## 1. Introduction

Interest in the human gut microbiota and its potential impact on human health has significantly increased since the development of metagenomic technologies that allow for the deep sequencing of microbial genomes. Trillions of bacteria live in the gut microbiota of the human digestive system and are established as a very complex environment. Beginning at birth, a wide range of genetic, nutritional, and environmental variables influence the composition of the gut microbiome [1]. After the beginning of the 20th century, Metchnikoff proposed that the human gut microbiome plays a part in both health and sickness [2]. Subsequently, the introduction of new molecular technologies that emerged in metagenomics spread a new perspective on microbiota research. The so-called “microbiota revolution” enabled the discovery of crucial links between pathogenic diseases and gut microbes [3].

The gut microbiota is composed of bacteria (generally including *Firmicutes*, *Bacteroidetes*, *Actinobacteria*, *Proteobacteria*, *Fusobacteria*, and *Verrucomicrobia phila*), yeasts, and viruses [4,5,6]. The gut bacteria perform a variety of functions, including vitamin production, pathogen defense, immune response stimulation, metabolism regulation and drug absorption [6]. High taxonomic diversity, microbial gene richness, and a stable core microbiota are frequently observed in healthy microbiota communities [6]. However, the relative distribution of microorganisms varies between individuals, even within the same person. Nearly every element of the host can be impacted by the microbiota, and dysbiosis is linked to a wide range of illnesses. In particular, dysbiosis has been implicated in cardiovascular and respiratory diseases, inflammatory bowel diseases, liver disorders, a variety of neoplasms, and metabolic illnesses via the intensification of a chronic inflammatory state [7,8,9]. It has also been suggested that gut microbes play a part in preserving the homeostasis of the gut–brain axis [9]. It has been supposed that different molecular patterns enhance microbiota-associated pathogenic states. Recently, it was discovered that the microbiota produces small molecules called genotoxins. In particular, the family of indolimines generated by the *M. morganii* strains associated with IBD–colorectal cancer can enhance colon carcinogenesis in mice and increase intestinal permeability [10]. Other small-molecule metabolites of Gram-positive and Gram-negative bacteria, such as those from *Clostridium perfringens* and *Clostridium ramosum* strains, directly damage DNA and cause the expression of double-strand break markers—as well as cell-cycle arrest—in epithelial cells [10]. Alterations in intestinal microbiota can therefore be correlated to pathological states, both through a non-specific mechanism of chronic inflammation and through specific molecular patterns (Figure 1); therefore, it is essential to achieve the ability to both modulate complex microbial communities and target individual members within these communities.

The intestinal mucosa provides a selective, permeable barrier for nutrient absorption and protection from external factors. Pathogens, xenobiotics, and food can disrupt the intestinal barrier, while genetic and immune factors predispose individuals to gut barrier dysfunction. The gut microbiota participates in regulating the integrity and function of the intestinal barrier in a homeostatic balance in various ways: opposing colonization by pathogens, promoting the differentiation of regulatory T (Treg) cells (which induce tolerance to lumen antigens), and stimulating B cells to secrete immunoglobulin (Ig)A to avoid bacterial translocation. Commensals also convert dietary fiber into short-chain fatty acids (SCFAs), which protect the gut barrier in various ways, including by providing energy for colonocytes and stimulating the production of mucus, antimicrobial proteins, and Treg cells. Dysbiosis and chronic disruption of the gut barrier can lead to the translocation of microbial components, the activation of pro-inflammatory patterns, and the production of systemic, low-grade inflammation [11,12,13]. 

To date, a multitude of molecules with a modulatory effect on gut microbiota have been developed [14,15,16,17,18]. Most of these agents are prebiotics, probiotics, synbiotics, or postbiotics [14,15,16,17,18]. However, some molecules known for their antimicrobial effect have been shown to improve the balance of the gut microbiota [19]. Due to their properties, these agents could not be included in the previously mentioned therapeutic groups [19]. Nonetheless, rather than being antimicrobial, their activity may be classified as eubiotic [19]. The improved understanding of this host–microbiota relationship has allowed for the development of microbiota-based therapies such as bacteria modulation, fecal microbiota transplantation (FMT), and molecular techniques (phage therapy and engineered bacteria). However, despite great interest in their efficacy, most of these interventions are not available in clinical practice [20]. The purpose of this review is to examine new therapeutic perspectives for the modulation of microbiota, including molecules with a eubiotic effect, FMT, and molecular techniques. Given the enormous intra- and interpersonal variability of the human gut microbiota and its alterations in the context of related pathologies, it would be desirable to achieve a “personalization” in therapeutic modulation. The aim of this review is to analyze some of the most recent and innovative perspectives in the field of gut microbiota modulation in order to obtain such personalization.

## 2. Eubiotics: Drugs to Modulate the Gut Microbiota

Prebiotics and probiotics have historically played a crucial role in the regulation of the gut microbiota. Prebiotics are typically non-digestible food components that selectively encourage the growth and activity of a small number of bacteria in the digestive tract [14]. Probiotics are a group of advantageous microorganisms. Probiotics are live microorganisms which, when administered in adequate amounts, confer a health benefit to the host [15]. Synbiotics (a combination of prebiotics and probiotics) and postbiotics (inanimate microorganisms and/or their components that confer a health benefit to the host) have been recently introduced, showing a beneficial effect upon the dysbiotic state [16,17,18].

An overview of the conditions for which there is evidence that the oral administration of a specific prebiotic, probiotic, or synbiotic is effective is reported in Table 1. For a more comprehensive and detailed analysis, we recommend referring to The World Gastroenterology Organisation Global Guidelines 2023 [21].

Nevertheless, pharmacological agents not included in the above-mentioned groups demonstrate an interesting ability to alter microbiota, leading to dysbiosis, with a significant effect on associated diseases. Antimicrobials, natural compounds, and even metabolites of the same gut microbiota have been considered. A new group, eubiotics, has been proposed [54,55]. All these molecules have a somewhat antimicrobial activity; however, rather than determining a global depletion in bacteria abundance, the antimicrobial activity leads to a modulation of the composition of microbiota, which can be beneficial for the host [54,55].

### 2.1. Rifaximin: New Perspectives for an Old Antibiotic

During the last half of the 20th century, a number of new antibacterial agents came into clinical use, providing clinicians with a variety of options when treating many types of infectious diseases; among these options, antibiotics played a major role [56]. In addition to their beneficial effect against pathogenic bacteria, antibiotics are associated with significant short- and long-term alterations in the composition of the human microbiota [57]. The way an antibiotic affects the gut microbiota depends largely on its class, as well as the composition of the microbiota in the gut before the antibiotic is administered [58].

The use of an antibiotic results in a decrease in alpha (α) diversity, which measures the variety of bacterial taxa present in a given individual’s microbiota, and in beta (β)-diversity, which is a marker of the homogeneity of bacterial relative abundance [58,59]. Since suppressing susceptible microbes can create an ecological niche for opportunistic pathogenic bacteria that are resistant to the antibiotic, increasing the host’s susceptibility to post-antibiotic infection, the majority of studies focus on the dysbiotic effect of antibiotics [57,60,61]. Additionally, it has been demonstrated that antibiotics enrich phage-encoded genes, which transfer resistance to the administered drug and unrelated antibiotics and encourage interactions between phages and bacteria, thereby enhancing the exchange of resistance genes [62].

Nonetheless, some antibiotics have shown an intriguing function in the clinical modulation of the gut microbiota. Taking into account changes in bacterial abundance, the non-systemic antibiotic rifaximin, which has bactericidal and bacteriostatic activity against both aerobic and anaerobic bacterial species, has demonstrated promising results [63]. Rifaximin also has bile-acid-dependent solubility, which increases its effectiveness in the small intestine while inhibiting colonic bacteria only moderately [64]. Rifaximin irreversibly binds the bacterial DNA-dependent RNA polymerase, inhibiting bacterial protein synthesis [65]. To date, rifaximin is usually administered with beneficial effects in the management of diseases associated with an alteration in the gut microbiota, such as irritable bowel syndrome (IBS), diverticular disease, and hepatic encephalopathy (HE) [66,67,68]. Orally administered rifaximin determines minimal changes in the composition of gut microbiota, promoting the growth of bacterial species with a beneficial impact [63]. In addition, rifaximin modulates the inflammatory response by upregulating the expression of NF-kB via the pregnane X receptor [69,70] and downregulating the pro-inflammatory cytokines interleukin-1B and tumor necrosis factor alpha (TNFα) [71,72]. As observed in a clinical trial based on a multi-tagged pyrosequencing analysis, rifaximin modulates the networks among several bacteria (*Enterobacteriaceae*, *Bacteroidaceae*, *Veillonellaceae*, *Porphyromonadaceae*, and *Rikenellaceae*) and bacterial metabolites when administered to patients with mild HE [73]. An analysis of the serum of treated patients revealed an increase in saturated and unsaturated fatty acids [73]. Furthermore, rifaximin increased bacterial diversity, the *Bacteroidetes*/*Firmicutes* ratio, and the abundance of *Faecalibacterium prausnitzii*, a butyrate producer with potent anti-inflammatory properties, in a second smaller study involving 15 patients with IBS [74]. An analysis of gut microbiota diversity showed that the clinical response was associated with a slight increase in α-diversity [74]. Treatment with 1200 mg of rifaximin daily for 10 days increased the abundance of *Lactobacilli* in another report on 19 patients with various gastrointestinal and liver disorders (inflammatory bowel disorder, IBS, diverticular disease, and HE), with no effects upon the α-diversity of the gut microbiota [54]. A recent clinical trial on patients with HE demonstrated that daily treatment with 1200 mg of rifaximin increased the abundance of *Proteobacteria* while decreasing the abundance of *Firmicutes* [75]. A metagenomic analysis highlighted that a decrease in the prevalence of *Veillonella*, *Haemophilus*, *Streptococcus*, *Parabacteroides*, *Megamonas*, *Roseburia*, *Alistipes*, *Ruminococcus*, and *Lactobacillus* was also associated with rifaximin administration. Despite this trend of bacterial abundance modification, the stability of the gut microbiota was not affected by rifaximin administration, and its minimal antibacterial effect in the intestine was confirmed in this report [75]. While the overall composition of the gut microbiota is unaffected by the treatment, rifaximin causes a relative rather than an absolute change in the abundance of these helpful bacteria. This might be because some microbes that are sensitive to rifaximin experienced a minimal reduction in abundance that is not statistically significant while others, such as *Lactobacilli*, developed resistance [54]. Interestingly, Brigidi et al. showed a significant initial decrease in the fecal abundance of *Lactobacilli* in individuals with mild or severe ulcerative colitis who were receiving a higher dosage of rifaximin (i.e., 1800 mg daily). However, the abundance of *Lactobacilli* returned after three cycles of therapy that each lasted 10 days. Moreover, *Lactobacilli*, which were particularly susceptible to rifaximin at the beginning of the study, developed resistance to the drug (to a mean value of 12 μg/mL) [76].

In order to improve the pharmacological approach and incorporate it into therapeutic options, it is necessary to investigate the mechanisms that underlie the clinical impact exerted by this beneficial gut microbiota perturbation.

### 2.2. The Multi-Layered Mechanisms of Rifaximin

Rifaximin, as previously mentioned, differs from other antibiotics as it does not affect the gut microbiota’s overall composition. Indeed, rifaximin increases the abundance of beneficial gut bacteria and promotes metabolic modifications, balancing the relationship between the host and the bacteria with a well-recognized positive clinical effect.

Although the underlying biological mechanisms are not fully understood, some metabolic networks may be suggested. Rifaximin treatment protected mice exposed to malathion from stress-induced oxidative damage in a pre-clinical study [77]. In the gut microbiota of rifaximin-treated mice, acetate- and propionate-producing bacteria, such as *Lactobacillus* spp., *Bacteroides* spp., *Prevotella* spp., *Streptococcus* spp., *Phascolarctobacterium succinatutens*, and *Negativicutes* spp., were found in high concentrations [77]. Therefore, treatment with rifaximin effectively increased the production of short-chain fatty acids (SCFAs) by modulating the gut microbiota [78]. Dupraz et al. showed that intestinal gamma-delta (γδ) T cells in mice and humans produce less IL-17 and IL-22 when microbiota-produced SCFAs, particularly propionate, are present. This results in a decrease in the inflammatory state [79]. Furthermore, the administration of rifaximin was linked to an increase in the expression of peroxisome proliferator-activated receptor gamma coactivator-1 alpha, a key metabolic regulator whose expression is elicited by SCFAs [77,80]. As a result, SCFAs might be the means through which rifaximin reduces systemic inflammation and exerts its beneficial effect. Another preclinical model highlighted rifaximin’s function in controlling systemic inflammatory responses. By inhibiting the activation of the TLR-4/NF-B signaling pathway and downregulating inflammatory factors such as TNF-α, IL-6, IL-17A, and IL-23, rifaximin significantly diminished the severity of clinical disease in mice with proteoglycan-induced ankylosing spondylitis. In this instance, a microbiological analysis showed an increase in *Lactobacillus* and *Bacteroides*, with a higher *Bacteroidetes*/*Firmicutes* ratio [81]. Additionally, in vitro studies performed in human gut epithelial cells indicated that rifaximin decreases apoptosis and increases tight junction protein expression by activating the TLR4, MyD88, and NF-kB pathways [82]. Meanwhile, increased populations of bacteria such as *Lactobacilli* have been shown to reduce the production of pro-inflammatory cytokines and TNF-α to prevent the spread of pathogenic bacteria and to modulate intestinal permeability [83,84,85].

As a result, rifaximin not only creates an anti-inflammatory environment by regulating intestinal metabolic pathways but also helps to reestablish the function of the intestinal barrier by reducing intestinal permeability, preventing the translocation and overgrowth of bacteria that could be harmful to the individual [55].

Patel et al. proposed that these preclinical implications could be transferred to a human model in a recent randomized controlled clinical trial [86]. In comparison to the placebo group, 90 days of rifaximin treatment decreased the plasma levels of tumor necrosis factor alpha (TNF-α) and circulating neutrophil TLR-4 expression in patients with cirrhosis and hepatic encephalopathy [86]. Additionally, metagenomic quantification techniques showed that rifaximin reduced the levels of sialidase-rich species that degrade mucin (i.e., *Streptococcus* spp., *Veillonella atypica* and *parvula*, *Akkermansia*, and *Hungatella*), preventing the so-called “oralization” of the gut microbiota [86]. These bacteria are present in the oral microbiota, but they are also more prevalent in the gut microbiota in conditions such as cirrhosis [87]. Sialidase alters the intestinal permeability by degrading O-glycans in the gut mucin barrier. Rifaximin also promoted an intestinal microenvironment rich in TNF-α and interleukin-17E (IL-17) [86]. It has been demonstrated that IL-17E (also known as IL-25), unlike other isoforms of IL-17, is implicated in the integrity of the gut barrier [88,89]. Intestinal tuft cells produce IL-17E in response to indole-acetic propionic acid, a tryptophan metabolite involved in gut homeostasis and anti-inflammatory pathways mediated by IL-10 [88,89]. It is also intriguing to note that plasma TNF-α levels decreased while intestinal TNF-α increased, demonstrating how the maintenance of gut homeostasis and the control of systemic inflammation can be achieved by increasing an inflammatory cytokine in a specific microenvironment. Clinically, rifaximin treatment resulted in a recovery of neurocognitive function, in addition to the complete resolution of hepatic encephalopathy. Additionally, rifaximin allowed for the preservation of beta (β) diversity in the gut microbiota, whereas the placebo group saw a decrease in this metric [86].

Rifaximin demonstrates great potential for maintaining the balance between the host and the gut microbiota. Molecular and microbiological data are strongly consistent with clinical evidence. Rifaximin’s effectiveness, however, may be impacted by various disruptions to the gut homeostasis that are related to various pathological conditions. On the other hand, rifaximin is a versatile drug because it has the ability to restore balance instead of acting on a single target. Regarding one of the main issues with antibiotic therapy, rifaximin showed no long-term effects, and its impact on bacterial resistance appears to be absolutely negligible [75]. The eubiotic effects of rifaximin are summarized in Figure 2.

Rifaximin therapy is associated with the modulation of the gut microbiota, which increases the production of short-chain fatty acids (SCFAs) and reduces the stimulation of the Toll-like receptor 4 (TLR4) pathway via pathogen-associated molecular patterns (PAMPs). This leads to a decrease in the production of pro-inflammatory cytokines such as IL17A, IL22, TNF-α, and IL-6 and an increase in the production of anti-inflammatory cytokines such asIL10, and IL17E. 

### 2.3. Other Antimicrobial Agents: Triclosan

Triclosan is a nonionic broad-spectrum antimicrobial agent. Its chemical name is 2,4,4′-trichloro-2′-hydroxydiphenyl ether. It is widely used in toothpaste, food storage containers, medical products, personal care products, and plastic cutting boards [90]. Triclosan acts as a detergent, directly disrupting the integrity of the bacterial membrane [91]. Moreover, it interferes with the synthesis of bacterial fatty acids by inhibiting the enoyl-acyl carrier protein (enoyl-ACP) reductase [92]. At low concentrations, it limits bacterial growth, while at high concentrations, it is bactericidal [93].

Since its introduction, numerous studies have examined the effects of triclosan on living organisms to determine its safety. A randomized study of triclosan-containing household and personal care products carried out to evaluate the safety of triclosan showed that triclosan exposure did not induce global reconstruction or the loss of microbial diversity in the gut microbiota [94]. Nevertheless, because triclosan-containing products included toothpaste, the oral microbiota was also examined. So far, even in sites where triclosan was applied directly, no significant differences in α-diversity were detected [94]. Triclosan seems to not have an antibacterial effect at the dosage used in household and personal care products, despite the fact that this effect may be present at higher concentrations and for longer periods of exposure [94,95].

Another preclinical model sought to define perturbations to the gut microbiota caused by triclosan. Microbiota composition was evaluated at three, twenty-one, and fifty-two weeks after the administration of a low dose of triclosan. Following exposure to triclosan (50 mg/kg/day), the abundance of *Bacteroidetes* increased at week 52. At the same dose after 52 weeks, triclosan slightly (but not significantly) reduced the abundance of *Firmicutes*. At 21 and 52 weeks of age, triclosan decreased the levels of *Akkermansia muciniphila* at the species level. Low doses of triclosan increased α-diversity after three weeks when compared to the control group [96]. Triclosan resistance mechanisms include changes in the enoyl-ACP reductase and efflux pumps. However, despite the widespread use of products containing triclosan, the community’s overall resistance and cross-resistance rates are low [97].

Triclosan consequently gained attention as a modulator of the gut microbiota, and it was considered as a drug for the treatment of dysbiosis in a recent preclinical study [98]. Mice received a continuous high-fat diet for 20 weeks and were then treated with triclosan at a dose of 400 mg/kg/d for the final 8 weeks [98]. In mice fed a high-fat diet, triclosan increased the ratio of *Bacteroidetes*/*Firmicutes* and decreased the number of pathogenic Gram-negative bacteria, such as *Helicobacter*, *Erysipelatoclostridium*, and *Citrobacter*, as shown by a metagenomic analysis [98]. Triclosan also increased the relative abundance of *Lactobacillus*, *Bifidobacterium*, and *Lachnospiraceae*, which protect against abnormal metabolic processes [98]. The increase in α- and β-diversity also demonstrated triclosan’s improvement of bacterial diversity and richness [98].

On the other hand, a cohort study showed that triclosan exposure causes a decrease in the diversity of the gut microbiota in breast-fed infants [99]. This study, however, is not representative of the general population due to the small sample size and the high vulnerability of the infant gut microbiota to antimicrobial agents. Additionally, rather than thinking of triclosan as a perturbator of a balanced microenvironment, it would be interesting to consider it as a modulator of the gut microbiota in patients with a state of gut dysbiosis [99]. Indeed, triclosan is employed in periodontal disease, which could be considered a model of oral dysbiosis [100]. Nonetheless, oral dysbiosis, which is characterized by an increased abundance of *Porphyromonas gingivalis* and *Aggregatibacter actinomycetemcomitans*, has been associated with the impairment of the gut microbiota equilibrium and the loss of gut microbial diversity [101,102]. Moreover, these oral pathobionts have been associated with NAFLD and NASH progression in preclinical models [101,103]. As a consequence, periodontal therapy can improve oral and gut dysbiosis in patients with chronic liver diseases [104]. In particular, periodontal therapy in cirrhotic individuals reduces blood levels of inflammatory cytokines and lipopolysaccharide (LPS), with beneficial effects upon both quality of life and in mitigating the development of hepatic encephalopathy [104]. Triclosan, as a treatment for periodontal disease [105], has been shown to reduce the production of pro-inflammatory cytokines (such as IL-8, IL-1 α, and TNFα) in a human epithelial cells, monocytes, and fibroblast cultures exposed to LPS [106]. More specifically, triclosan treatment inhibits the TLR-4 pathway by inducing the microRNA miR146a to downregulate IRAK1 and TRAF6 proteins. Conversely, triclosan exposure increases the epithelial cells’ production of other bioactive anti-microbial molecules, such as β-dsefensins [106]. Interestingly, β-defensins have been proposed as a key factor in the physiological homeostasis between the host and the microbiome [107].

In conclusion, triclosan, an antimicrobial agent with a long history, has been suggested to have a positive impact on the microbiota in the human gut. In addition to its well-known antibacterial action, triclosan could have a pleiotropic effect on different cell types, directly orchestrating physiological homeostasis between microbiota and their host. However, more investigations are necessary.

Triclosan’s eubiotic effects are summarized in Figure 3.

Some studies suggest that triclosan may have a beneficial modulatory effect on the gut microbiota, promoting bacterial diversity and the abundance of “good” bacteria. In vitro studies have shown that triclosan can promote the epithelial–microbiota balance by inducing the production of β-defensins and inhibiting the activation of the Toll-like receptor 4 (TLR4) pathway. Furthermore, triclosan treatment is associated with a decrease in the production of inflammatory cytokines by immune cells. 

### 2.4. Natural Products: Promising Agents for the Modulation of Microbiota

Evodiamine is an alkaloid, mainly present in the Evodia Fructus, which is officially listed in the Chinese Pharmacopoeia as a remedy for patients suffering from viral hepatitis, cholangitis, and gastric ulcers, among other disorders [108]. Evodiamine has been demonstrated to have an antimicrobial activity in vitro, inhibiting bacterial topoisomerase I [108].

Evodiamine’s impact on intestinal inflammation and the gut microbiota was examined in a preclinical study [109]. In order to create a intestinal inflammatory tumor mouse model, azomethane/sodium dextran sulfate was used [109]. The tumor model was then treated with evodiamine and 5-aminosalicylic acid. Evodiamine and 5-aminosalicylic acid both prevented the growth of tumors and induced the death of tumor cells [109]. A quantitative polymerase chain reaction analysis showed that evodiamine and 5-aminosalicylic acid reduced the number of *Enterococcus faecalis* and *Escherichia coli* while increasing the abundance of *Bifidobacterium*, *Campylobacter*, and *Lactobacillus* when compared to the control group [109]. The IL6/STAT3/P65 signaling pathway was inhibited, and levels of inflammatory factor, d-lactic acid, and serum endotoxin were all significantly decreased in the evodiamine group [109]. Evodiamine also showed efficacy in preventing colorectal tumors in a mouse model of chemically induced colitis [110]. In this model, evodiamine increased the abundance of SCFA-producing bacteria, inhibiting the harmful bacteria [110]. After evodiamine treatment, histological analysis showed a remarkable reversion of intestinal epithelial structure destruction, inflammatory cells infiltration, and crypt loss. The intestinal barrier was also restored, with an increased expression of occludin, zonula occludens-1, and E-cadherin [110,111]. Additionally, evodiamine decreased the expression of pro-inflammatory genes involved in the Wnt signaling pathway, the Hippo signaling pathway, and the IL-17 signaling pathway [110]. This anti-inflammatory effect could be mediated by the downregulation of the nuclear factor-kappa B (NF-κB) signal and the inhibition of NLRP3 inflammasome activation, as shown in another sodium-dextran-sulfate-induced colitis murine model [111].

The effect of evodiamine was also evaluated on *H. pylori* in an in vitro gastric adenocarcinoma model [112]. Evodiamine treatment decreased the bacterial production of the type IV secretion system components and the system subunit protein A protein and the expression of cytotoxin-associated antigen A (CagA) and vacuolating cytotoxin A (VacA). This resulted in a reduction in CagA end VacA proteins in tumor cells. In addition, evodiamine specifically prevented the *H. pylori*-infection-induced stimulation of signaling proteins such as NF-κB and the mitogen-activated protein kinase (MAPK) pathway. As a result, IL-8 secretion in tumoral cells was reduced [112].

Evodiamine and berberine were combined in order to determine their effect on the gut microbiota of rats fed a high-fat diet [113]. The treatment with evodiamine and berberine increased the diversity of the gut microbiota globally [113]. The gut microbiota profiles were determined via the high-throughput sequencing of the bacterial 16S ribosomal RNA gene [113]. In comparison to the control group, there were higher proportions of SCFA-producing bacteria (*Lactobacillus*, *Prevotella*, *Ruminococcaceae*, and *Bacteroides*). In contrast, the key bacteria responsible for the imbalance in the gut microbiota in the group with a high-fat diet were *Fusobacteria* and *Lachnospiraceae* [113]. The high-fat-diet group’s *Firmicutes*/*Bacteroidetes* ratio was significantly higher than that of the control group [113]. The administration of evodiamine and berberine reverted the increasing trend of the ratio. Evodiamine–berberine also lowered the intestinal submucosal edema typical of a high-fat diet and the mucosal inflammatory cell infiltration [113]. As a result, the rats treated with evodiamine and berberine showed a significant reduction in body weight, as well as plasma triglyceride and total cholesterol levels [113]. Liver injury (measured via plasma levels of aspartate aminotransferase, alanine aminotransferase, and gamma-glutamyl transpeptidase) was significantly reduced [113]. Evodiamine’s eubiotic effects are summarized in Figure 4.

The capacity of other substances to alter the microbiota has been researched in relation to their potential role in the treatment of colorectal cancer. Natural products, and in particular natural extracts, are frequently the source of anticancer agents. In mice with heterotopic xenograft colorectal cancer, an ethanol extract of *Euphorbia lathyris* was found to reduce tumor size [114]. When compared to mice without colorectal cancer, a gut microbiota analysis showed a significant decrease in the abundance of *Lactobacilli*, with predominance of colitogenic bacteria such as *Akkermansia* and *Turicibacter*. *Turicibacter* disappeared after treatment with an ethanol extract of *Euphorbia lathyris*, and Lactobacillus abundance recovered [114]. Nevertheless, it is still unclear whether the *Euphorbia lathyris* extract impacted the microbiota directly or whether it modulated gut microorganisms as a result of its anti-neoplastic effects.

Propolis is a resinous material collected by bees from the buds and resins of plants and mixed with bee enzymes, pollen, and wax [115]. It has been used as a traditional remedy worldwide. In some countries, it is considered a complementary medicine, while it is regulated as a food supplement in others [116]. It contains a variety of molecular compounds, such as flavonoids, terpenes, phenolic acids, β-steroids, and the derivatives of sesquiterpenes, naphthalene, and stilbenes [117]. Its antioxidant, anti-inflammatory, and antimicrobial activities are probably due to its polyphenol contents. Concerning the microbiota, propolis has been demonstrated to modulate intestinal bacteria, resulting in an improvement in intestinal barrier function in diabetic rats [117]. However, due to the content variability of propolis, studies require the standardization of its content. A recent preclinical trial showed that a standardized polyphenol propolis mixture modulated the gut microbiota in an in vitro human model, determining an increase in the production of SCFAs [116]. A propolis mix with 7.21 g of total polyphenols/g was specifically used. The propolis extract’s polyphenols included quercetin, apigenin, pinobanskin, chrysin, pinocembrin, and galangin at defined dosages. The standardized polyphenol combination underwent an in vitro digestion process which mimicked human digestion before being fermented by fecal microbiota isolated from healthy adults and children, obese children, celiac children, and children with a food allergy. The production of SCFAs significantly increased following these processes, as shown by a combined chromatographic and UV detection technique. Particularly, samples from the allergic, obese, and celiac patients had significantly higher levels of acetic acid production than the samples from healthy people. On the other hand, compared to other samples, the samples from individuals with allergies had higher levels of propionic acid [116].

Polyphenols are primarily produced by plants as defense against pathogenic microorganisms [118]. As different microbes have different metabolisms, polyphenols appear to selectively inhibit bacterial growth. Tannic acid, for instance, did not affect the growth of *Bifidobacterium infantis* and *Lactobacillus acidophilus*, but it inhibited the growth of *Clostridium clostridiiforme*, *E. coli* ATCC 25,922, and *Enterobacter cloacae* ATCC 13,047 in anaerobic conditions. Tannic acid is a potent iron chelator, and its ability to inhibit bacterial growth may be due to the fact that it deprives bacteria of the iron that they require for metabolism (*Bifidobacterium infantis* and *Lactobacillus acidophilus* do not) [119]. Other polyphenols have a more specific mechanism of action. Epigallocatechin directly binds the peptidoglycan in the *S. aureus* cellular wall, causing osmotic damage and bacterial death [120]. Furthermore, tea polyphenols exert direct damage on the outer and inner membranes of *S. marcescens*, dramatically increasing cellular permeability and thus inhibiting bacterial growth [121].

One of the most investigated polyphenols is resveratrol, a phytoalexin present in many plants such as grapes, peanuts, and berries [122]. Resveratrol has demonstrated antimicrobial and antifungal activity by preventing the formation of biofilms [123,124,125]. Resveratrol, previously administered to mice receiving intrarectal treatment with the oxidizing agent 2,4,6-trinitrobenzenesulfonic acid, demonstrates a beneficial effect on the gut microbiota. Genomic DNA analysis and 16S rRNA gene sequencing revealed that oxidative stress increased species such as *Bacteroides acidifaciens* and decreased species such as *Ruminococcus gnavus* and *Akkermansia muciniphila*, according to an analysis of stool microbiota. The administration of resveratrol returned the gut bacteria to their homeostatic levels and increased the production of butyric acid. A mesenteric lymph node analysis via cytometry showed a significant increase in the percentages of both anti-inflammatory CD4+FOXP3+ Tregs and CD4+IL10+ in the resveratrol-treated group. Meanwhile, inflammatory Th1/Th17 cells were suppressed, and mucosal inflammation was also significantly reduced [126]. A microarray analysis of microRNA profiles in mesenteric lymph node cells highlighted that resveratrol treatment was associated with miR-31, Let-7a, and miR-132 downregulation [127]. All these microRNAs target anti-inflammatory T cells. In particular, miR-31 inhibits the production of FoxP3 [128]. It is noteworthy to observe that miR-31 expression in tissues connected to the disease is considerably greater among individuals with ulcerative colitis [127].

Moreover, by lowering inflammation and regulating hepatic lipid metabolism, resveratrol has been demonstrated to lower the risk of non-alcoholic fatty liver disease (NAFLD). Resveratrol treatment reduced hepatic steatosis in male mice fed with a high-fat diet and modified the composition of the gut bacteria by enhancing the growth of SCFA-producing bacteria such as *Allobaculum*, *Bacteroides*, and *Blautia* [129]. A clinical trial investigated the impact of resveratrol on the fecal microbiota of overweight men and women [130]. In comparison to women, men have a higher fecal abundance of *Bacteroidetes*. Resveratrol (282 and 80 mg/day, respectively) and another phenolic compound called epigallocatechin-3-gallate were combined for 12 weeks [130]. Men experienced a decline in the relative abundance of *Bacteroidetes*, but women did not. Administration had no effect on firmicutes, actinobacteria, gammaproteobacteria, *Akkermansia muciniphila*, sulfate-reducing bacteria, or acetogenic bacteria in either men or women. By using indirect calorimetry, it was demonstrated that fat oxidation was increased in men compared to women [130]. These findings highlight the importance of a gut microbiota modulator’s ability to restore an altered equilibrium. Instead of acting against a specific target, a good modulator suppresses a deregulated element until it resumes its homeostatic role [130]. Resveratrol’s eubiotic effects are summarized in Figure 4.

The Mediterranean diet, with a plant-based profile, is distinguished by a notable richness in polyphenols [131]. Indeed, a diet similar to the Mediterranean diet was strongly linked to a microbial composition in which SCFA-producing bacteria, such as *Ruminococcus*, predominated. When compared to other dietary patterns, it was also linked to lower levels of fecal calprotectin [132]. The majority of research on polyphenols use purified and more concentrated polyphenol extracts [133]. However, it has been shown that foods rich in polyphenols, such as almonds and cranberries, can modulate the gut microbiota as well [134,135].

Other relevant substances have been discovered to be elevated in the serum of people eating a Mediterranean diet. Indole-3-propionic acid is one specific example. As previously mentioned, this tryptophan metabolite, which is produced by gut bacteria, induces tuft cells to produce IL-17E, which has a modulatory effect on the gut microbiota, preventing lipoperoxidation and oxidative stress injury and reducing the synthesis of proinflammatory cytokines [136]. The administration of indole-3-propionic acid resulted in a significantly different composition of the intestinal microbiota in a sepsis mouse model, with an enrichment of the *Bifidobacteriaceae* family and a depletion of the *Enterobacteriaceae* family. In comparison to the control group, it resulted in lower serum inflammatory mediator levels and a higher survival rate [137]. This emphasizes once more how a compound that affects the gut microbiota can modify a systemic condition. In a different preclinical study, indole-3-propionic acid prevented the development of nonalcoholic steatohepatitis in mice fed a high-fat diet [138]. An analysis of the fecal microbiota via DNA sequencing techniques revealed a decrease in the *Firmicutes*/*Bacteroidetes* ratio as well as a decrease in the overall population of the pathobiont *Streptococcus* [138]. Rats fed a high-fat diet showed a loss of the normal villus structure of the ileum epithelium at the histological level. Treatment with indole-3-propionic acid improved the expression of tight junction proteins and restored the height of the ileum villus [138]. In addition, endotoxin plasma levels were lower in mice treated with indole-3-propionic acid than in the control group. As a result, when indole-3-propionic acid was administered, a significant histological reduction in steatohepatitis was seen [138]. Nonetheless, indole-3-propionic acid has been found to reverse dysbiosis induced by total abdominal irradiation in a mouse model, which was characterized by a decrease in the relative abundance of Lactobacillus and an increase in *Bacteroides acidifaciens* and *Ruminococcus gauvreauii* [139].

Evodiamine administration in murine models of chemically induced colitis has confirmed its potential role as a eubiotic. In fact, the modulatory effect of evodiamine results in an increase in α-diversity and a consequent inhibition of inflammation, as demonstrated by the decrease in IL17 production and NLRP3 inflammasome inhibition. Moreover, evodiamine can improve the epithelial barrier by restoring the expression of occludin, ZO-1, and E-cadherin after chemical damage. Resveratrol has also been shown to improve disruption to the gut microbiota after oxidative stress damage. The eubiotic effects of resveratrol are associated with an increase in the production of short-chain fatty acids (SCFAs) and the activation of the anti-inflammatory phenotype regulatory T cells (Tregs). 

The dynamic actions of pharmaceutical agents are a fascinating topic. Monodirectional approaches are insufficient when dealing with the complex microenvironment that is the microbiota. It is necessary to think about an intervention that can restore the disrupted balance, enhancing the resilience of the gut microenvironment. Far from being considered merely antimicrobial, these drugs are contributing to the definition of a new therapeutic paradigm. Single-target therapies have the risk of worsening the already fragile equilibrium. To restore the organism’s balance, rather than eliminating the bacteria thought to be responsible for the disruption, it is reasonable to promote the harmonious growth of all bacterial species, including the “harmful” ones. In this regard, a strategy that promotes a healthy metabolic and immunologic cooperation with the host, providing a quantitative and qualitative harmonic balance of the gut microbial components, is required. This is known as eubiosis because it is an ideal condition for a living system. Eubiotics refer to the strategies that can be used to achieve this goal. Antibiotics, but also chemically produced antimicrobials, plant extracts, or compounds and even metabolites of the same gut microbiota, could play a role. Thus, the pharmacological approach, along with dietary changes and lifestyle intervention, should be integrated into a complex model, leaving the door open to other techniques, among which the possibility of directly changing the composition of the microbiota is playing an increasingly intriguing role. A summary of the main eubiotics studied and their mechanisms of action is reported in Table 2.

## 3. Fecal Microbiota Transplantation

Fecal microbiota transplantation (FMT) is defined as the infusion of feces from a healthy, screened donor to a recipient to restore a disorder associated with a perturbation of the gut microbiota (dysbiosis). Over time, FMT has been proven to be an effective and safe procedure and has been established as a reliable treatment option for recurrent *Clostridioides difficile* infection (CDI) [140,141,142,143,144].

The FMT process is characterized by several key steps that may differ among different centers. The first cornerstone of the FMT framework is represented by the selection of stool donors [145]. Current guidelines for FMT in clinical practice [146] recommend a specific four-step selection process, including a clinical interview, blood and stool testing, a further questionnaire, and a direct molecular stool testing on the day of each donation [147]. In the last year, donor screening has been updated to prevent the risk of COVID-19 dissemination [148,149]. Recently, the need to make FMT an equitable, accessible, widespread, and secure procedure have led to the development of stool banks [150], with the principle aim of providing FMT to health centers in a safe and traceable manner. FMT protocols may also differ according to the route of delivery. More specifically, FMT could be administered by upper GI routes (including upper GI endoscopy and nasogastric/nasoduodenal tube) [151,152], capsules [153,154], and via lower GI routes (enema [155] and colonoscopy [142,156]). Colonoscopy and capsule administration are known to be the most effective routes in the treatment of recurrent CDI, with a cure rate of nearly 90% for colonoscopies [157] and 85% for capsules [158,159].

At present, and over the course of the years, FMT has been established as a reliable therapeutic alternative to vancomycin and fidaxomicin for the treatment of recurrent *Clostridioides difficile* infection (CDI) [140,141,142], as recommended by international guidelines and consensus reports [146,150,160]. In this setting, a large body of evidence supports the efficacy of FMT also in the treatment of severe and complicated CDI. Particularly, FMT has been proven to be effective in reducing the risk of CDI- associated bloodstream infections [161] and in decreasing the need for surgical treatment [162], with an increase of overall survival in patients with recurrent CDI [161]. It was also shown to be a cost-effective strategy for this disorder. FMT has also proven to be highly effective and superior to the standard of care alone, vancomycin, in achieving a sustained resolution of the first episode of CDI [163]. After the satisfactory results obtained in CDI, in recent years, a growing number of studies have been carried out to investigate the role of FMT in the treatment of noncommunicable chronic disorders, including inflammatory bowel disease (IBD) [164,165], irritable bowel syndrome (IBS) [152], psychiatric disorders [166], metabolic disorders [167], liver disease [168], autoimmune disorders, and cancer [169,170,171]. In these areas, FMT has achieved promising but heterogenous results, making it impossible to draw any firm clinical conclusions as to its efficacy in these settings.

Increasing evidence suggests that FMT’s clinical success in chronic disorders may be influenced by several factors related to the microbial characteristics of both donors and recipients or closely related to FMT working protocols. Recently, the role of ecological parameters and the taxonomic composition of the donor microbiome, including bacteria, bacteriophages, and fungi, has been investigated in several studies. In FMT trials [172,173] enrolling patients with IBD, the clinical response to FMT was very closely associated with higher donor evenness and richness. This finding correlates with the known evidence that alpha diversity can be considered a marker of human health and is indirectly related to the engraftment success [174,175,176]. Beyond ecological parameters, the compositional characteristics of the donor microbiome also appear to influence FMT efficacy. In a recent randomized trial [152] of patients with IBS, the use of a “super donor” was associated with a significantly higher success rate than a placebo. Moreover, the route of delivery and the amount of infused feces also appear to be other parameters able to influence clinical response after FMT. The clinical success of FMT is higher when the administration is performed via colonoscopy [157] or capsules [153,154], with the use of a suitable amount of feces (>50 g) [152] or with a protocol employing multiple infusions [142,177]. Furthermore, pre-conditioning with antibiotics may improve FMT’s clinical efficacy, probably modifying alpha diversity and improving engraftment. Unfortunately, nowadays, the interpretation of the reproducibility and effectiveness of FMT are fragmented by the adoption of different protocols.

The recent understanding of FMT’s mechanisms has allowed us to associate the role of donor microbiome engraftment (defined as the number of engrafting strains by the total detected strains in the donor and recipient [178]) with the clinical response [151,178,179]. The close association of engraftment with the clinical response to FMT has been identified in a recent metagenomic metanalysis of 24 FMT trials, including communicable and non-communicable disorders, in which engraftment was found to be higher in communicable disorders and after the antibiotic pre-conditioning of the recipient [178]. The increasingly crucial role of engraftment in FMT has created the need to assess it using precise tools, mainly whole-genome sequencing (WGS), which provides a higher taxonomic resolution than 16S rRNA gene sequencing. Despite its enormous value, a widespread diffusion of WGS is still limited by its high costs and analytical complexity which require staff with bioinformatics and computational skills [180].

Beyond the knowledge of FMT’s mechanisms for improving clinical response, another relevant issue in this field is represented by the precision and reproducibility of FMT, as the more we need targeted donor microbiomes, the more it will be hard to find and, in particular, keep them (as the donor microbiome can be perturbed over time). In recent years, novel, advanced FMT preparations [153,154]—including lyophilized fecal material, bacterial consortia, and live biotherapeutic products (LBPs) [181,182]—have emerged as alternatives to “classical” FMT and are also appearing in clinical practice for the prevention of CDI recurrence; however, their efficacy for noncommunicable disorders has to be proved yet.

## 4. Engineered Bacteria and Phage Therapy

One of the most significant limits of the current techniques used to modulate the gut microbiota is due to a lack of tools that can precisely modulate specific members of complex microbial communities. To the date, two different approaches have been hypothesized to overcome these limits: the first approach is based on delivering ex vivo-engineered bacteria into the gut, and the alternative approach involves taking advantage of the rich microbial ecosystem in the gut by genetically modifying the microbiome in situ through use of engineered bacteriophages [183].

### 4.1. Engineered Bacteria

The “ex vivo” approach provides bacteria that are engineered ex vivo to secrete therapeutic molecules or to sense one or more biomarkers and can be introduced to the microbiome. Engineered gut bacteria can be divided into three main classes: drug factory probiotics, diagnostic gut bacteria, and smart probiotics [184] (Table 3).

Drug factory probiotics are bacteria engineered to constitutively produce a therapeutic molecule within the body [184]. In 2000, a natural probiotic, *Lactococcus lactis,* was genetically modified to constitutively secrete the human anti-inflammatory cytokine protein interleukin-10 (IL-10); when orally administrated in rat models of colitis, a significant reduction of inflammation was assessed [185]. This evidence inspired countless subsequent studies: the same natural probiotic, *L. lactis*, was modified to secrete a variety of anti-inflammatory molecules, including anti-tumor necrosis factor (TNF) nanobodies (certolizumab) [186], IL-27 [187] and recombinant mouse heme oxygenase-1 (rmHO-1) [188]. All these studies used drug factory probiotics, which were administered orally, and they all showed that this method reduced inflammation in the mouse gut more than systemically administered drugs. These probiotics have been studied in a wide range of disorders: for example, *Lactobacillus gasseri*, which secreted full-length protein GLP1, induced the differentiation of rat epithelial cells into functional glucose-responsive insulin-producing cells, and this improved glucose control in a rat model of diabetes mellitus [189];

On the other hand, diagnostic gut bacteria are defined as bacteria that sense one or more biomarkers, compute that those biomarkers are present in a combination indicative of disease, and produce a reporter that can be externally measured by a clinician [184]. Using engineered bacteria to sense transient molecules that are degraded, modified, or absorbed before exiting the gut and thus cannot be easily captured and quantified by traditional non-invasive tests, is an exciting prospect for measuring novel biomarkers. In 2015, an engineered *E. coli Nissle* 1917 was used as diagnostic tool to detect the presence of liver metastasis in mouse models: the probiotic was engineered to express an enzyme that could cleave a systemically administered substrate, leading to a color change that was detected in urine. After oral delivery, this modified *E. coli Nissle* 1917 generated a high-contrast urine signal through the selective expansion of the probiotic in liver metastases, demonstrating that probiotics can be programmed to safely and selectively deliver synthetic gene circuits to diseased tissue microenvironments in vivo [190]. In 2017 [191], Daffler and his collaborators hypothesized that thiosulfate could serve as a novel biomarker of colitis and engineered *E. coli Nissle* 1917 to carry sensors that detected increased thiosulfate levels during chemically induced colitis in mice. Furthermore, the greater the extent of inflammation, the greater the thiosulfate receptor activity was, suggesting that thiosulfate may be a novel colitis biomarker.

Smart probiotics are bacteria that sense one or more biomarkers, compute that those biomarkers are present in a combination indicative of disease, and respond by delivering a precise dose of one or more appropriate therapeutics at the diseased tissue [184]. To date, such smart probiotics still are not available: further progress in synthetic biology is needed to create such a complex model of engineered bacteria. Selecting the chassis for smart probiotics requires careful evaluation: the capacity to survive transit through the gastrointestinal tract, colonize macroscopic or microscopic geographical locations within the gut, and reach desired densities there must be considered. Most engineered probiotics use a small set of bacterial species; *Lactococcus lactis* and *Escherichia coli* are commonly used for oral delivery to the gut; attenuated versions of *Salmonella enterica subsp. enterica serovar Typhimurium*, a pathogen capable of protein expression and immune stimulation in the human body, target the hypoxic tumor environment when administered systemically and the gut mucosa when provided orally [192,193]; attenuated *Listeria monocytogenes*, another pathogen, activates the immune system through growth within circulating immune cells to elicit antitumor responses [194].

There are several drawbacks that have limited this bacterial “cell therapy” approach. First, achieving stable engraftment of engineered microbes into the endogenous microbiota can be difficult. For example, the bacteria used in the clinic to date either have short or no colonization capacity in humans (*L. lactis* and *E. coli*) or are capable of being cleared by routine antibiotic administration (*S. Typhimurium* and *L. monocytogenes*). Moreover, there will likely be substantial variations in engraftment efficiency between different individuals due to inherent variations in their existing microbiomes. Thus, it is difficult to control or predict the long-term engraftment of exogenously delivered bacteria within any given patient population. Finally, even if the introduced strains can achieve stable engraftment, they may outcompete endogenous commensals and adversely alter the balance of the microbial ecosystem [183]. Furthermore, another concern is the uncontrolled growth of engineered bacteria within the gut. In order to reach the safe use of smart probiotics in clinical practice, biocontainment strategies are needed.

### 4.2. Phage Therapy

In addition to this “ex vivo” strategy, an “in vivo” approach has been developed and is one of the most cutting-edge frontiers in the manipulation of gut microbiota: phage therapy. Phages have been widely studied as a possible therapy against bacterial infection [195,196]; however, the aim of this review is to focus exclusively on their potential use as modulators of the gut microbiota. In the past decade, phages have been used for the precision editing of the gut microbiota. In 2017, phages against adherent-invasive *E. coli* (a bacteria implicated in the pathogenesis of inflammatory bowel disease) were administered in mice models of dextran-sodium-sulfate-induced colitis: mice receiving phage treatment were found to be protected from DSS-induced colitis, and *E. coli* colonization was reduced [197]. A phase I/IIa randomized, double-blind, placebo-controlled clinical trial is underway to assess the safety and efficacy of the oral administration of phages that target intestinal adherent–invasive *E. coli* in patients with Crohn’s disease in remission (NCT03808103) [198]. *Enterococcus faecalis* is significantly increased in patients with alcoholic hepatitis compared with patients without alcohol use disorder and in particular, the presence of *E. faecalis* strains that produce cytolysin (a bacterial exotoxin) correlates with worse outcomes: in 2019, a study showed that mouse models treated via oral gavage with phages specifically targeting cytolysin-positive *E. faecalis* showed reductions in ethanol-induced liver disease [199]. These are great examples that highlight the potential of phage therapy in the modulation of gut microbiota, but both these studies make use of natural phages: the next step in the field of phage therapy is represented by the possibility of engineering phages to deliver recombinant DNA to target bacteria or deliver drugs to a specific location with specificity down to the strain level.

In situ microbiome engineering involves the delivery of transgenes into specific members of the endogenous microbiota. Since it directly modifies the endogenous microbiota, this type of therapy overcomes all the major drawbacks of the “ex vivo” strategy: by editing species that have long established long-term colonization, the genetically altered bacteria will likely be retained in the gut for much longer than the exogenously dosed bacteria, possibly achieving stable engraftment. Additionally, since the microbiome’s composition is not directly altered, there is probably less of an effect on other microbiome members that could endanger the delicate balance of this microbial ecosystem [183]. Although engineered bacteriophages represent an interesting and promising perspective in the field of gut microbiota modulation, current clinical trials involving phages are limited to their use as antimicrobials or diagnostics [183]. This is due to several difficulties that have yet to be resolved as of the time of writing, including the need to modulate bacterial target specificity, overcome bacterial immunity, and limit the spread of any exogenous DNA and engineered organisms using appropriate biocontainment systems. Phages’ exceptional specificity (often only a few strains of a given bacterial species are targeted by these viruses, and even closely related strains of the same bacterial species cannot be infected by them) [200,201,202] is indeed an advantage when a specific pathogen must be eliminated without disturbing other members of the same genus or species, but it can be an obstacle when the objective is to broadly affect multiple members of the microbiome among multiple individuals: to achieve a therapeutic effect, phage specificity must be adjusted to a level that permits the transduction of a sufficient number of bacterial strains [201].

Engineering the phage’s host recognition domain (HDR), a virus structure that controls the interface between phages and bacterial surface molecules, can achieve this objective [203]: several investigators have attempted to rationally design phages with an altered host range through HRD engineering [204,205,206,207,208]. Moreover, to achieve a stable modification in the gut microbiota though the use of phages, overcoming bacterial innate immunity is indispensable. Of all the known bacterial phage defense systems, the one called restriction modification (R-M) is the most abundant and is found in 75% to 95% of all known bacterial genomes [209,210,211]. This system functions through two components: endonucleases, which recognize and cleave specific DNA sequences, and cognate methyltransferases, which protect host DNA of the same sequence via methylation. In 2015, Robert et al. created a database called REBASE for the components of R-M systems. Covering recognition and cleavage sites of both restriction enzymes and methyltransferases of all completely sequenced genomes, REBASE is a powerful tool for designing plasmids with the proper methylation patterns necessary for known bacterial targets [212].

Overall, phage-based therapies could become promising and powerful approaches for modulating the gut microbiota, but more basic and preclinical studies, as well as properly designed randomized, double-blind, placebo-controlled trials, are required to help the field move forward.

## 5. Conclusions

Collectively, a growing suite of strategies for manipulating the gut microbiome—both highly targeted, and whole-ecosystem-based—have recently appeared in mainstream medicine. Eubiotics, mainly rifaximin, are increasingly becoming a cornerstone of specific disorders, including IBS and HE. Novel eubiotics, including other antimicrobials (e.g., triclosan) and natural products (e.g., polyphenols and propolis), appear to be a promising approach, but further studies are needed to confirm their value as microbiome modulators. FMT, after becoming a well-established therapy against recurrent CDI, has been investigated in other non-communicable disorders with variable results. To improve the outcomes of FMT in disorders beyond CDI, an understanding of the therapeutic mechanisms (e.g., microbial engraftment), as well as the application of novel technologies (e.g., whole genome sequencing), is advocated. Finally, novel approaches, e.g., the use of engineered probiotic bacteria or a bacteriophage-based therapy, have recently appeared as promising opportunities for providing the targeted and tailored therapeutic modulation of gut microbiota, but their role in clinical practice has yet to be clarified.

## Figures and Tables

**Figure 1 antibiotics-12-00868-f001:**
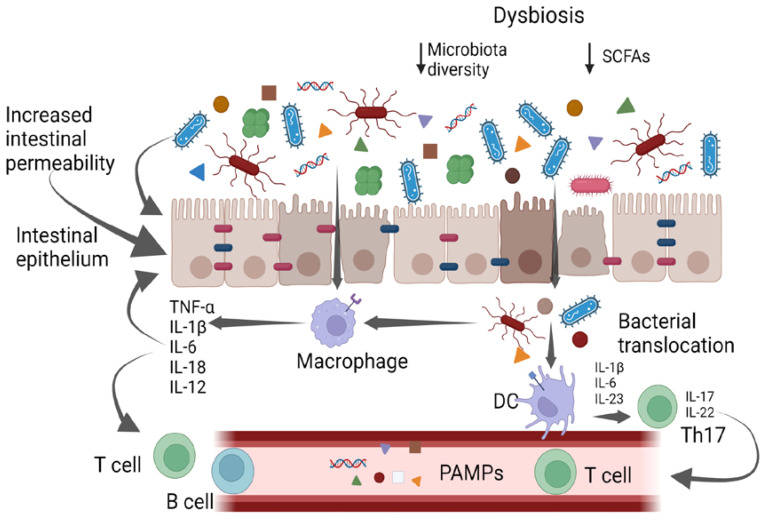
Gut bacteria and systemic inflammation. PAMPs—pathogen-associated molecular patterns; DCs—dendritic cells. Created with BioRender.com.

**Figure 2 antibiotics-12-00868-f002:**
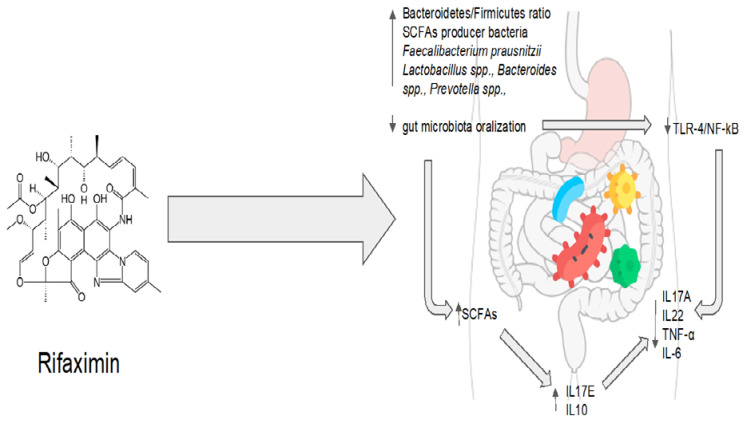
Rifaximin’s eubiotic effects. IL17A: interleukin 17A; IL22: interleukin 22; TNF-α: tumor necrosis factor alpha; IL-6: interleukin 6; IL10: interleukin 10; IL17E: interleukin 17E.

**Figure 3 antibiotics-12-00868-f003:**
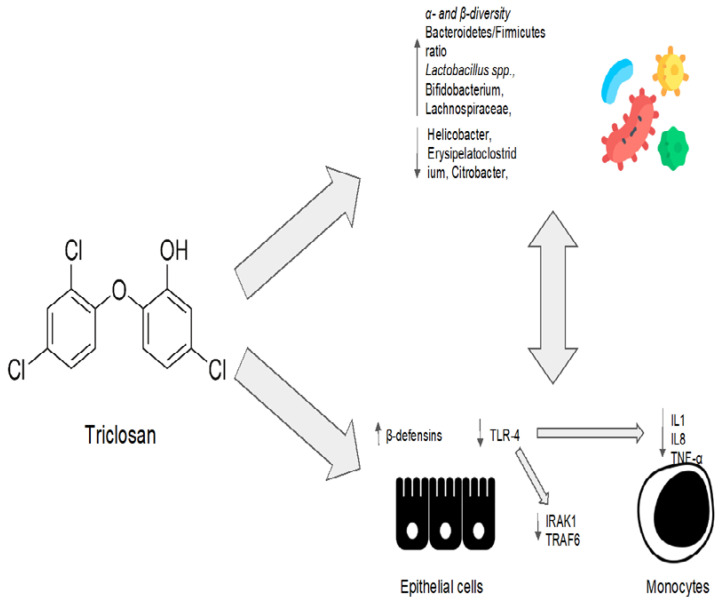
Triclosan’s eubiotic effects. IL1: interleukin 1; IL8: interleukin 8; TNF-α: tumor necrosis factor alpha; IRAK1: interleukin 1 receptor-associated kinase 1; TRAF6: TNF-receptor-associated factor 6.

**Figure 4 antibiotics-12-00868-f004:**
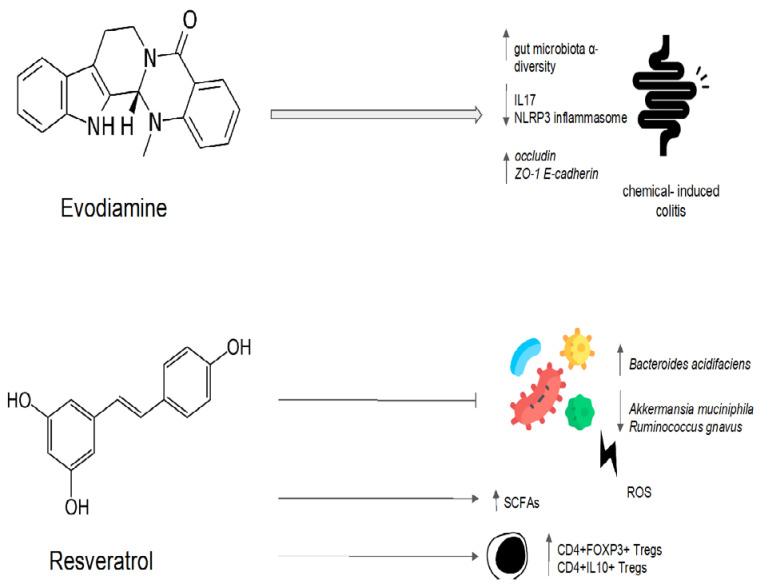
Possible mechanisms of the eubiotic effect of evodiamine and resveratrol. IL17: interleuchina-17; NLRP3: NOD-like receptor family, pyrin domain-containing protein 3; ROS: reactive oxygen species; ZO-1: zonula occludens-1; E-cadherin: epithelial cadherin.

**Table 1 antibiotics-12-00868-t001:** Overview of clinical studies on probiotics, prebiotics, synbiotics in particular diseases.

Probiotic Strain/Prebiotic/Synbiotic	Disorder
- *Lactobacillus paracasei* B 21060 or *L. rhamnosus* GG [22] - *Saccharomyces boulardii* CNCM I-745 [23] - *Enterococcus faecium* SF68 [24]	Treatment of acute diarrhea in adults
- Yogurt with *L. casei* DN114, *L. bulgaricus*, and *Streptococcus thermophilus* [25,26] - *Lactobacillus acidophilus* CL1285 and *L. casei* (Bio-K+ CL1285) [25,26]	Antibiotic-associated diarrhea (AAD)
- *Lactobacillus acidophilus* CL1285 and *L. casei* LBC80R [27,28,29] - Yogurt with *L. casei* DN114 and *L. bulgaricus*, and *Streptococcus thermophilus* [27,28,29]	Prevention of *Clostridioides-difficile*–associated diarrhea (or the prevention of recurrence)
- *Lactobacillus rhamnosus* GG [30] - *Bifidobacterium animalis* subsp. lactis Bb12, *Lactobacillus rhamnosus* GG [31]	Coadjuvant therapy for *Helicobacter pylori* eradication
- Mixture containing strains of *Lactobacillus plantarum*, *L. casei*, *L. acidophilus*, *L. delbrueckii* subsp. *bulgaricus*, *Bifidobacterium infantis*, *B. longum*, *B. breve*, and *Streptococcus salivarius* subsp. *Thermophilus* [32,33,34] - *Lactobacillus acidophilus* plus *Bifidobacterium bifidum* [33,34,35]	Prevention of diarrhea associated with radiotherapy
- Shen Jia fiber plus *Bifidobacterium* and *Lactobacillus* in tablets [36]	Prevention of diarrhea associated with enteral nutrition
- Lactulose [37] - Mixture containing strains of *L. plantarum*, *L. casei*, *L. acidophilus*, *L. delbrueckii* subsp. *bulgaricus*, *Bifidobacterium infantis*, *B. longum*, *B. breve*, and *Streptococcus salivarius* subsp. *Thermophilus* [38,39]	Hepatic encephalopathy
- *Lactobacillus casei, L. rhamnosus, Streptococcus thermophilus, Bifidobacterium breve, L. acidophilus, B. longum*, and *L. bulgaricus*, plus fructooligosaccharide [40]	Non-alcoholic fatty liver disease (NAFLD)
- *Bifidobacterium bifidum* MIMBb75 [41,42] - *Lactobacillus plantarum* 299v (DSM 9843) [43,44]	Irritable bowel syndrome (IBS)
- *Bifidobacterium bifidum* (KCTC 12199BP), *B. lactis* (KCTC 11904BP), *B. longum* (KCTC 12200BP), *Lactobacillus acidophilus* (KCTC 11906BP), *L. rhamnosus* (KCTC 12202BP), and *Streptococcus thermophilus* (KCTC 11870BP) [45] - *Lactobacillus reuteri* DSM 17938 [46]	Functional constipation
- *Lactobacillus casei* subsp. DG [47]	Uncomplicated symptomatic diverticular disease
- *Lactobacillus casei* strain Shirota in fermented milk [48]	Small-bowel injury due to non-steroidal anti-inflammatory drugs (NSAIDs)
- Mixture containing strains of *Lactobacillus plantarum, L. casei, L. acidophilus, L. delbrueckii* subsp. *bulgaricus, Bifidobacterium infantis, B. longum, B. breve*, and *Streptococcus salivarius* subsp. *thermophilus* [49,50] - Mixture containing strains of *Lactobacillus plantarum, L. casei, L. acidophilus, L. delbrueckii* subsp. *bulgaricus, Bifidobacterium infantis, B. longum, B. breve*, and *Streptococcus salivarius* subsp. *thermophilus* [49]	Pouchitis
- Mixture containing strains of *Lactobacillus plantarum, L. casei, L. acidophilus, L. delbrueckii* subsp. *bulgaricus, Bifidobacterium infantis, B. longum, B. breve*, and *Streptococcus salivarius* subsp. *thermophilus* [51] - *Escherichia coli Nissle* 1917 [52,53]	Ulcerative colitis

**Table 2 antibiotics-12-00868-t002:** Eubiotics, mechanisms of action, and a shift in microbial composition.

	Pathology	Intervention	Microbiota Modifications	Mechanism
Rifaximin	Hepatic encephalopathy	Several clinical trials with oral rifaximin administration in the setting of HE [73,74,75,76]	Increase in α-diversity Increase in *Bacteroidetes*/*Firmicutes* ratio Increase in abundance of *Faecalibacterium prausnitzii* Decrease in the prevalence of *Veillonella*, *Haemophilus*, *Streptococcus*, *Parabacteroides*, *Megamonas*, *Roseburia*, *Alistipes*, *Ruminococcus*, and *Lactobacillus* was also associated with rifaximin administration	Changes the gut microbiota, promoting the growth of bacterial species with a beneficial impact. Modulation of inflammatory response by upregulating NF-kB expression via the pregnane X receptor and downregulating pro-inflammatory cytokines interleukin-1B and tumor necrosis factor-alpha (TNFα).
Triclosan	No clinical indications to date. It is widely used in toothpaste, food storage containers, medical products, personal care products, and plastic cutting boards.	Preclinical model: the composition of the microbiota was evaluated at three, twenty-one, and fifty-two weeks after low-dose triclosan administration [96]	Increase in the abundance of *Bacteroidetes*. Slightly (but not significantly) reduced the abundance of *Firmicutes*. Decreased the levels of *Akkermansia muciniphila* at the species level. Low doses of triclosan increased α-diversity after three weeks when compared to the control group.	Acts as a detergent, directly disrupting the integrity of the bacterial membrane. Interferes with the synthesis of bacterial fatty acids by inhibiting the enoylacyl carrier protein (enoyl-ACP) reductase. As a periodontal disease treatment, inhibits the TLR-4 pathway by inducing microRNA miR146a to downregulate IRAK1 and TRAF6 proteins. Increases epithelial cells’ production of other bioactive anti-microbial molecules such as β-defensins.
Evodiamine	No clinical indications todate.	Preclinical study: mouse intestinal inflammatory tumor model treated with evodiamine and 5-aminosalicylic acid [109]. Preclinical study: Evodiamine efficacy in preventing colorectal tumors in a chemical-induced colitis mice model [110]. Treatment of *H. pylori* in an in vitro gastric adenocarcinoma model [112].	Reduction in abundance of *Enterococcus faecalis* and *Escherichia coli*. Increase in abundance of *Bifidobacterium*, *Campylobacter*, and *Lactobacillus*. Increase in the abundance of SCFA-producing bacteria, inhibiting the harmful bacteria. Decrease in the bacterial production of CagA end VacA proteins into tumor cells.	The IL6/STAT3/P65 signaling pathway was inhibited, and levels of inflammatory factors, d-lactic acid, and serum endotoxin were all significantly decreased in the evodiamine group. Increase in the expression of occludin, zonula occludens-1, and E-cadherin. Decrease in the expression of pro-inflammatory genes involved in the Wnt signaling pathway, the Hippo signaling pathway, and the IL-17 signaling pathway. Evodiamine specifically prevented *H. pylori*-infection-induced stimulation of signaling proteins such as NF-κB and the mitogen-activated protein kinase (MAPK) pathway. As a result, IL-8 secretion in tumoral cells was reduced.
Propolis	No clinical indications to date; regulated as a food supplement.	Preclinical trial: propolis mix with 7.21 g of total polyphenols/g orally administred to investigate changes in microbiota composition [116].	Increase in the production of SCFAs.	Unknown
Resveratrol	No clinical indications to date.	Preclinical trial: previously administered to mice receiving intrarectal treatment with the oxidizing agent 2,4,6-trinitrobenzenesulfonic acid [126].	Increased in *Bacteroides acidifaciens* abundance. Decrease in abundance of *Ruminococcus gnavus* and *Akkermansia muciniphia*.	Increase in the percentages of both anti-inflammatory CD4+FOXP3+ Tregs and CD4+IL10+. Suppression of inflammatory Th1/Th17. Down-regolation of microRNA miR-31, Let-7a, and miR-132 in mesenteric lymph node cells. All these microRNAs target anti-inflammatory T cells. In particular, miR-31 inhibits the production of FoxP3 [128].

**Table 3 antibiotics-12-00868-t003:** Classification and main studies of engineered bacteria.

	Definition	Main Studies
Drug factory probiotic	Bacteria engineered to constitutively produce a therapeutic molecule within the body [184]	*L. lactis* modified to constitutively secrete IL-10 in mouse model of colitis [185]. *L. lactis* modified to constitutively secrete certolizumab (anti-TNF) in mouse model of colitis [186]. *L. lactis* modified to constitutively secrete IL-27 in mouse model of colitis [187]. *L. lactis* modified to constitutively secrete rmHO-1 in mouse model of colitis [188]. *L. gasseri* modified to constitutively secrete GLP1 in mouse model of diabetes [189].
Diagnostic gut bacteria	Bacteria that sense one or more biomarkers, compute that those biomarkers are present in a combination indicative of disease, and produce a reporter which can be externally measured by a clinician [184].	*E. coli Nissle* 1917 as diagnostic tool to detect liver metastasis in mouse models. [190]. *E. coli Nissle* 1917 to sense thiosulfate levels in mouse chemically induced colitis [191].
Smart probiotics	Bacteria that sense one or more biomarkers, compute that those biomarkers are present in a combination indicative of disease, and respond by delivering a precise dose of one or more appropriate therapeutics at the diseased tissue [184].	Not available to date.

## Data Availability

Not applicable.

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
