# Peer review of "Future Modulation of Gut Microbiota: From Eubiotics to FMT, Engineered Bacteria, and Phage Therapy"

_antibiotics, 2023, doi:10.3390/antibiotics12050868_

Round 1

Reviewer 1 Report

Airola et al have worked on summarizing the tools available and currently in use to regulate gut microbiome. They have done a great job in including all the relevant literature related to FMT and phage therapy. This review was very well written and up to date regarding the literature in the field. The authors were mindful to include the limitations of the tools mentioned as well. The subject of the manuscript is significant for the field and has a lot of scope in creating an impact.

There are a few opportunities to improve the clarity and impact of the manuscript.

-          Throughout the manuscript these are various spots where words like “in vivo”, “in vitro”, “E. coli” etc are not italicized. Please be mindful of these.

Page 3 Line 85-91 – Please add relevant references to add value to this statement.

Page 4 Line 131-134 - Do the authors mean bacterial diversity in gut microbiome? Because the overall diversity would be affected. As said in a couple of sentences above rifaximin increases bacterial diversity and abundance of good bacteria.

Page 4 Line 143-145 - Is there literature that co-relates the concentration with developing resistance or increasing abundance of species.

Page 5 Line 221-223 - Add some information about whether this study showed the antimicrobial effect of triclosan containing products in physiological conditions.

Page 6 Line 231- Does this contribute to antimicrobial resistance by any chance?

Page 8 Line 364-367 - In general, could the authors touch base on whether the concentrations of NP in diet/vegetables/fruits enough to cause this change or supplements are needed?

Author Response

Rome, 18 April 2023

Dear Reviewer,

We would like to thank you for the careful assessment of our paper and for your precious comments, that improved the quality of our paper.

We have done our best to address comments satisfactorily and hope that you will appreciate the revised version of the paper. Please find below a point-by-point response to your comments.

Best regards

Gianluca Ianiro, on behalf of all co-authors

POINT-BY-POINT RESPONSE

Throughout the manuscript these are various spots where words like “in vivo”, “in vitro”, “E. coli” etc are not italicized. Please be mindful of these.

R: We have italicized latin words as requested

Page 3 Line 85-91 – Please add relevant references to add value to this statement.

R: We have added references as requested

Page 4 Line 131-134 - Do the authors mean bacterial diversity in gut microbiome? Because the overall diversity would be affected. As said in a couple of sentences above rifaximin increases bacterial diversity and abundance of good bacteria.

R: We thank the reviewer for this insight, and have changed the text accordingly

Page 4 Line 143-145 - Is there literature that co-relates the concentration with developing resistance or increasing abundance of species.

R: We have reported a study that co-relates higher concentration of rifaximin with bacteria resistance. It seems to have an effect also on bacteria abundance.

Page 5 Line 221-223 - Add some information about whether this study showed the antimicrobial effect of triclosan containing products in physiological conditions.

R: We have provided more information about the antimicrobial effect of triclosan containing products in physiological conditions according to this study.

Page 6 Line 231- Does this contribute to antimicrobial resistance by any chance?

R: We have discussed triclosan resistance and cross-resistance mechanisms and their relationship with triclosan use as microbiota modulator.

Page 8 Line 364-367 - In general, could the authors touch base on whether the concentrations of NP in diet/vegetables/fruits enough to cause this change or supplements are needed

R: We have discussed in more detail the usage of natural product extracts, as they typically contain larger quantities of active compounds than fruits or vegetables. We have added information on the influence of dietary intake with food on microbiota, particularly of polyphenols.

Reviewer 2 Report

The review article Future modulation of gut microbiota: from eubiotics to FMT, engineered bacteria, and phage therapy shows that many bacteria, yeasts, and viruses inhabit the human intestine. Dynamic balance among microorganisms is associated with the well-being of the human being, but several disorders can cause the loss of balance of this microbial flora. Therefore, it is necessary to seek treatment to restore the balance of the microbiota. So this review shows several ways to maintain and achieve the balance between all these microorganisms in the human intestine. Therefore, it is an article of great importance and scientific impact.

My observations:

On page 7, on lines 301, 302, and 303, the author cites a preclinical study, but without details, I suggest putting more details on this study, which used propolis extract

Author Response

Rome, 18 April 2023

Dear Reviewer,

We would like to thank you for the careful assessment of our paper and for your precious comments, that improved the quality of our paper.

We have done our best to address comments satisfactorily and hope that you will appreciate the revised version of the paper. Please find below a point-by-point response to your comments.

Best regards

Gianluca Ianiro, on behalf of all co-authors

POINT-BY-POINT RESPONSE

On page 7, on lines 301, 302, and 303, the author cites a preclinical study, but without details, I suggest putting more details on this study, which used propolis extract.

R: We have added more details on the study matter of the comment, providing more information about the propoli extract composition and its effect on individuals differing on age and health conditions.

Reviewer 3 Report

Dear authors, your manuscript is good, and you have put in your efforts very nicely. I would like to add some work to this manuscript to make it in better shape. 

1. Abstract is not well written; it is not representing the text story nicely.

2. Most of the content is written superficially, authors should have to pay more attention to adding the mechanism of the probiotics and their molecular relation.

3. Authors should have to add one good figure that has the potential to explain the gut bacteria and their correlation with the molecular mechanism of the diseases. 

4. Authors should have to add one table of different probiotics, prebiotics, synbiotics, and their related benefits in the particular disease. 

Thank you

Author Response

Rome, 18 April 2023

Dear Reviewer,

We would like to thank you for the careful assessment of our paper and for your precious comments, that improved the quality of our paper.

We have done our best to address comments satisfactorily and hope that you will appreciate the revised version of the paper. Please find below a point-by-point response to your comments.

Best regards

Gianluca Ianiro, on behalf of all co-authors

POINT-BY-POINT RESPONSE

The manuscript has undergone a language editing by a native speaker (Dr. Benjamin H. Mullish, who has been added among authors), to solve all grammatical issues

Abstract is not well written; it is not representing the text story nicely.

R: We have modified the abstract, emphasizing the role of eubiotics in gut microbiota modulation and highlighting the limits of current techniques used to modulate gut microbiota, and then the need of new techniques to provide a targeted and tailored therapeutic modulation of gut microbiota, such as the use of engineered bacteria and bacteriophages.

Most of the content is written superficially, authors should have to pay more attention to adding the mechanism of the probiotics and their molecular relation.

R: We have explained with more precision the mechanism of action of different microbiome modulators discussed in this review article. In particular, for triclosan, evodiamine, and polyphenols we have disentangled the main antimicrobial mechanism and we have described in more detail their pleiotropic effect, focusing on the relationship between microbiota modulation and molecular and cellular pathways with a beneficial effect for the host.

Authors should have to add one good figure that has the potential to explain the gut bacteria and their correlation with the molecular mechanism of the diseases.

R: We have added a new figure (Figure 1) in order to explain the correlation between alterations in the gut microbiota and mechanism of the diseases.

Authors should have to add one table of different probiotics, prebiotics, synbiotics, and their related benefits in the particular disease.

R: We have added a new table (Table 1) with an overview of the conditions for which there is evidence that oral administration of a specific prebiotic, probiotic, or synbiotic is effective.

Reviewer 4 Report

To,

The Chief Editor,

Antibiotics, MDPI,

Manuscript ID: antibiotics-2331096

Subject: Submission of comments on the manuscript in “Antibiotics"

Dear Chief Editor Antibiotics, MDPI,

Thank you very much for the invitation to consider a potential reviewer for the manuscript (ID: antibiotics-2331096). My comments responses are furnished below as per each reviewer’s comments. 

 In the reviewed manuscript, the authors summarized the most recently introduced innovations in the field of therapeutic microbiome modulations. In general, the manuscript represents a very big piece of information in a logical presentation. however, in my opinion, the MS needs major revisions. I have some suggestions to improve this manuscript:

  1. I have read the entire manuscript and my initial comment is that manuscript is well-written. The topic is relevant and interesting.
  2. The structure of the abstract should be improved and please give more precise objectives here (such as in the Abstract).
  3. Introduction grammatical issues appear to be most prevalent in the introduction, making for very confusing reading. Further, the introduction is short but has no clear thread.
  4. The figures quality is not up to the standards. Higher-resolution versions will be needed for publication, for example, in Figures 1, further, figure texts are not readable.
  5. References: shall have to correct the whole References according to the ”Instructions for the Authors”, e.g. the Journal and scientific name must be in italics, the year must be bold and you shall have to use the abbreviated Journals name.
  6. The conclusion section is very lengthy. The author should emphasize this in a better way.
  7. Thank you and best wishes

Author Response

Rome, 18 April 2023

Dear Reviewer,

We would like to thank you for the careful assessment of our paper and for your precious comments, that improved the quality of our paper.

We have done our best to address comments satisfactorily and hope that you will appreciate the revised version of the paper. Please find below a point-by-point response to your comments.

Best regards

Gianluca Ianiro, on behalf of all co-authors

POINT-BY-POINT RESPONSE

The manuscript has undergone a language editing by a native speaker (Dr. Benjamin H. Mullish, who has been added among authors), to solve all grammatical issues.

The structure of the abstract should be improved and please give more precise objectives here.

R: We have modified the abstract, emphasizing the role of eubiotics in gut microbiota modulation and highlighting the limits of current techniques used to modulate gut microbiota, and then the need of new techniques to provide a targeted and tailored therapeutic modulation of gut microbiota, such as the use of engineered bacteria and bacteriophages.

Introduction grammatical issues appear to be most prevalent in the introduction, making for very confusing reading. Further, the introduction is short but has no clear thread.

R: All the review has undergone a language editing by a native speaker, to solve all the grammatical issues. We have modified the introduction, emphasizing the correlation between gut microbiota alterations and pathological states, then the importance of gut microbiota modulation

The figures quality is not up to the standards. Higher-resolution versions will be needed for publication, for example, in Figures 1, further, figure texts are not readable.

R: We have provided a high-resolution version of the image and changed the size of texts.

References: shall have to correct the whole References according to the ”Instructions for the Authors”, e.g. the Journal and scientific name must be in italics, the year must be bold and you shall have to use the abbreviated Journals name.

R: We have modified the References according the “Instruction for the Authors”

The conclusion section is very lengthy. The author should emphasize this in a better way.

R: We have modified the conclusion section in order to be more synthetic and effective.

Round 2

Author Response

Rome, 03 May 2023

Dear Reviewer,

We would like to thank you for the careful assessment of our paper and for your precious comments, that improved the quality of our paper.

We have done our best to address comments satisfactorily and hope that you will appreciate the revised version of the paper. Please find below a point-by-point response to your comments.

Best regards

Gianluca Ianiro, on behalf of all co-authors

POINT-BY-POINT RESPONSE

Authors highlighted the importance of the diverse microbiota and their contribution. Although authors have tried to cover some latest finding related to the microbial communities related to therapeutic modulations. Every section must have corresponding pictorial depictions for better understanding and to make the article more attractive for readers.

R: We have added several pictorial depictions.

Metagenome analysis-based data should be included which showing community profile/abundance of microorganism related to the aim of the study.

R: We have included metagenome analysis-based data.

Also, the focus of manuscript, one figure must be present showing their structural components, bonds, etc.

R: We have added pictorial depictions showing main eubiotics mechanisms of action

Authors can provide graphical abstract, if possible, for a better understanding of the manuscript

R: Thank you for your suggestion, we will consider providing a graphical abstract if allowed by the Journal rules.

Introduction: written well. Need restructure with define research gaps and objectives of this review article.

R: We restructured the introduction in order to make objectives appearing clearer

Some long sections don't have any references cited.

R: we have added properly references

Name of the bacteria must be in Italics.

R: We have bacteria’s name words as requested

First, the figure is poor, secondly, the full design is not any novelty

R: As we added other figures in the current version of the manuscript, we are keen on removing Figure 2.

A table must be included which emphasize the importance, procedures and molecular significance of the gut bacteria and the therapeutic bonding.

R: We have added a table as requested.

Reviewer 4 Report

Dear Chief Editor,

Thank you for providing the opportunity to review the revised manuscript. The manuscript is improved considerably after revision according to the reviewer's comment. Now this study is a suitable contribution to the IJMS. I recommend the manuscript for publication.

Thank you

With best regards

Author Response

Rome, 03 May 2023

Dear Reviewer,

We would like to thank you for the careful assessment of our paper and for your precious comments, that improved the quality of our paper.

Best regards

Gianluca Ianiro, on behalf of all co-authors